# Reinforcement Learning for Locally Checkable Labeling Problems

## Abstract

We address the challenge of solving locally checkable labeling (LCL) problems on graphs using machine learning. Unlike prior supervised approaches that depend on ground-truth algorithms or enforce unique solutions, we propose a reinforcement learning framework that requires only verifiers to evaluate correctness. This formulation allows models to learn solution strategies independently, without bias toward specific algorithmic procedures, and inherently supports the discovery of non-unique solutions. We evaluate our method on four fundamental LCL problems, demonstrating its ability to generalize effectively, outperform supervised baselines, and provide a versatile foundation for learning algorithmic reasoning on graphs.

## Introduction

Graph algorithms play a central role in solving problems across diverse domains, including network optimization, resource management, and data organization (Newman 2010). Many such algorithms are designed to assign discrete labels to graph elements, such as nodes or edges, based on well-defined rules. Examples include finding maximal independent sets, minimal vertex covers, or edge colorings. A particularly intriguing subset of these tasks is locally checkable labeling (LCL) problems, where solutions can be verified using localized checks on small subgraphs.

The algorithms used to solve these problems typically operate through step-wise procedures involving discrete state transitions. For instance, Luby's algorithm (Luby 1985) iteratively builds a maximal independent set by selecting nodes based on local rules and updating their states. Learning to replicate such algorithmic behavior poses significant challenges, as it requires bridging the gap between the continuous representations of machine learning models and the discrete nature of algorithmic solutions. Recent work, such as GraphFSA (Grötschla et al. 2024) or Discrete Neural Algorithmic Reasoning (Rodionov and Prokhorenkova 2024), have addressed this challenge by incentivizing discrete transitions during the training of a continuous system. However, these methods often struggle with more complex problems or rely on access to ground-truth labels and solutions, which limits their scalability and applicability.

In this work, we propose a reinforcement learning (RL) framework for solving LCL problems in a multi-agent set-

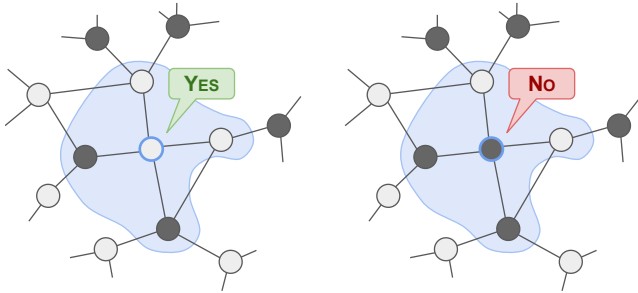

Figure 1: Finding a maximal independent set is locally checkable. A verifier can validate both conditions — the solution set has to be independent and maximal — for each node and its neighborhood. In this visualization, dark nodes represent nodes that are part of the solution set. Executing the verifier on the blue highlighted node entails checking all incident nodes and adjacent edges within the blue area. On the left, the conditions are met, while this is not the case on the right.

ting. In our approach, each agent, representing a node or edge, learns decision-making policies based solely on local observations. They are trained through problem-specific verifiers that evaluate solutions on localized neighborhoods for correctness. This verifier-driven approach removes the need for ground-truth labels or pre-defined algorithms, allowing the model to discover solutions independently of specific solving strategies. Additionally, unlike supervised learning methods, which often require unique solutions or external symmetry-breaking mechanisms, our framework naturally handles problems with multiple valid solutions. This flexibility broadens its applicability to a wider range of graph problems.

The paper introduces this RL-based framework and provides a practical implementation tailored to LCL problems. We evaluate the method on several fundamental graph problems, including maximal independent set and minimal vertex cover, and demonstrate its ability to generalize effectively across problem instances. Our experimental results showcase its potential to learn discrete algorithmic behav-

ior in graph-based tasks. These findings suggest that RL, coupled with local verifiers, offers a promising direction for addressing algorithmic graph problems with broader implications for learning and reasoning in discrete domains.

## Preliminaries

We present the key concepts required to understand our approach, with additional preliminaries and related work provided in the Appendix.

### The GraphFSA Framework

In GraphFSA, ach node executes a finite state automaton operating over a discrete set of states. More formally, the GraphFSA $\mathcal{F}$ consists of a tuple $(\mathcal{M}, \mathcal{Z}, \mathcal{A}, \mathcal{T})$. $\mathcal{F}$ is applied to a graph $G = (V, E)$ and consists of a set of states $\mathcal{M}$, an aggregation $\mathcal{A}$ and a transition function $\mathcal{T}$. At time $t$, each node $v \in V$ is in state $s_{v,t} \in \mathcal{M}$. In its most general form, the aggregation $\mathcal{A}$ maps the multiset of neighboring states to an aggregated value $a \in \mathcal{Z}$ of a finite domain.

$$a_{v,t} = \mathcal{A}(\{\{s_{u,t} \mid u \in N(v)\}\})$$

Here $\{\{\}\}$ denotes the multiset and $N(v)$ the neighbors of $v$ in $G$. At each timestep $t$, the transition function $\mathcal{T} : \mathcal{M} \times \mathcal{Z} \to \mathcal{M}$ takes the state of a node $s_{v,t}$ and its corresponding aggregated value $a_{v,t}$ and computes the state for the next timestep $s_{v,t+1} = \mathcal{T}(s_{v,t}, a_{v,t})$. Note that $\mathcal{Z}$ is modeled to be a finite domain.

**Aggregation functions.** The transition value $a$ for node $v$ at time $t$ is directly determined by aggregating the multi-set of states from all neighboring nodes at time $t$. The aggregation $\mathcal{A}$ specifies how the aggregated value is computed from this neighborhood information. Note that this formulation of the aggregation $\mathcal{A}$ allows for a general framework in which many different design choices can be made for a concrete class of GraphFSAs.

**Starting and final states.** The FSA uses starting states $S \subseteq \mathcal{M}$ to encode the discrete set of inputs to the graph problem. Final states $F \subseteq \mathcal{M}$ are used to represent the output of a problem. In node prediction tasks, we choose one final state per class. Opposed to other states, it is not possible to transition away from a final state, meaning that once such a state is reached, it will never change.

### Locally Checkable Labeling Problems

A graph problem is locally checkable if the correctness of a solution can be verified within all local neighborhoods. Locally checkable labeling (LCL) problems are typically node-centric and defined for graphs with bounded maximum degree (Balliu et al. 2019; Chang 2020, 2023; Balliu et al. 2020; Brandt et al. 2017). However, we relax this restriction to accommodate graphs with arbitrary degrees.

Formally, an LCL problem is a tuple $(\Sigma_{\text{in}}, \Sigma_{\text{out}}, r, \mathcal{C})$, where $\Sigma_{\text{in}}$ and $\Sigma_{\text{out}}$ are finite input and output label sets to make the graph attributed, $r$ is a constant radius, and $\mathcal{C}$ is a set of allowed $r$-hop neighborhoods. A solution is correct if every $r$-hop neighborhood matches a graph in $\mathcal{C}$ under isomorphism. For edge-centric tasks, labels and neighborhoods

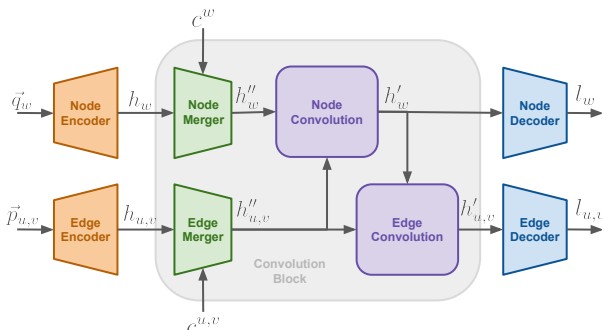

Figure 2: The proposed architecture consists of encoding MLPs, a convolution block, and decoding MLPs. Inside the convolution block, the cell values are first merged with extracted input embedding. Then a single node convolution layer performs neighborhood exchanges to propagate information without violating locality constraints. Before node and edge embeddings can be decoded to logits, the two endpoints' processed node state representations are aggregated across each edge.

are defined around edges rather than nodes. A verifier algorithm checks the local neighborhood and outputs YES if a neighborhood matches an element of $\mathcal{C}$, and NO otherwise. To accept a proposed solution, all neighborhoods have to output YES. Common LCL problems include finding maximal independent sets, vertex or edge colorings, and maximal matchings. An example of such a verifier is depicted in Figure 1.

## Problem Formulation

We follow the framework introduced by GraphFSA (Grötschla et al. 2024). We focus on problems defined on a graph structure $G = (V, E)$ that can be solved using a finite set of states $Q$ and $q_v$ denotes the state of node $v$. Further, we will concentrate on tasks that belong to the class of locally checkable labeling problems.

In contrast to the original formulation of GraphFSA, we relax the condition for the transition function. Namely, the aggregation function, which determines how $\mathcal{T}$ behaves, no longer has to be explicitly discretized to a finite domain. Instead, $\mathcal{T}$ directly maps a node's current state, and its neighbors' states to the node's next state. That is, the state gets updated according to

$$q^v \leftarrow \mathcal{T}(q^v, \{\{q^u \mid u \in N(v)\}\})$$

This formulation does not explicitly define the aggregation function, such as thresholded counting, and therefore allows for greater generality. In particular, it enables the incorporation of additional continuous inputs, such as random bits, into the computation of $\mathcal{T}$, which can facilitate tie-breaking. This flexibility is critical for handling input graphs that admit multiple valid solutions or require symmetry-breaking to resolve ambiguities, as illustrated in Figure 3. We maintain the concept of terminal states—states from which no further transitions are possible. Once all nodes

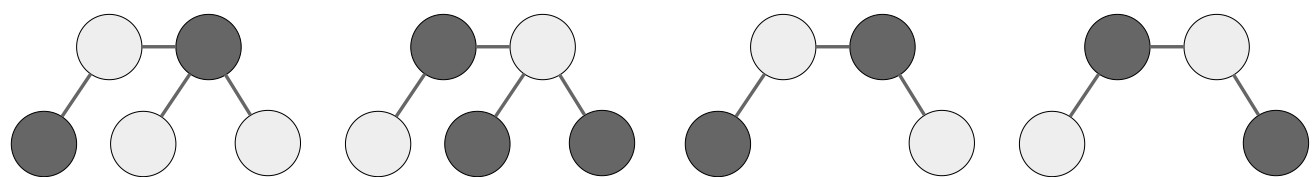

Figure 3: The solution for problems we consider does not need to be unique. For example, for MIS, the same input graph gives rise to two different solutions displayed above. Moreover, while for the two graphs on the left, one could choose one solution over the other based on the graph topology, the two examples on the right are completely symmetrical and, as such, require additional means of symmetry breaking to assign different output labels to nodes in the same orbit. This problem remains even when the GNN is fully expressive under 1-WL.

reach terminal states, the algorithm halts, ensuring a well-defined termination condition.

## Model Architecture

We introduce Verifier-based Algorithmic Reasoning using Reinforcement Learning (VARL) approach that uses a Graph Neural Network (GNN) to learn the actor policies that follows an encode-process-decode paradigm. At each timestep $t$, we are given the graph $G = (V, E)$, cell values, all node and edge states. For each node $w \in V$ and each edge $\{u, v\} \in E$ our network outputs action logits to be taken to transition to timestep $t + 1$. A schematic representation of the architecture can be seen in Figure 2.

First, two Multi-Layer Perceptrons (MLP) encode the one-hot encoding of the node states $\vec{q}_w$ and edge states $\vec{p}_{u,v}$ into a $d$-dimensional embedding $h$.

$$h_w = \text{MLP}_{\theta_1}(\vec{q}_w) \qquad h_{u,v} = \text{MLP}_{\psi_1}(\vec{p}_{u,v})$$

The processing is done with a convolution block that first combines node and edge embedding together with their respective cell values into $h''$.

$$h''_w = \text{LIN}_{\theta_2}(h_w || c^w) \qquad h''_{u,v} = \text{LIN}_{\psi_2}(h_{u,v} || c^{u,v})$$

Afterwards, it locally propagates the information among node and edge neighborhoods using a message-passing layer to derive the updated node and edge embedding $h'$. Any node convolution layer fits here. But as a default setting, we use a slightly modified Graph Isomorphism Network (GIN) (Xu et al. 2019) layer, including edge features to construct the node update. Furthermore, we use max aggregation, which was shown to be beneficial in algorithmic settings (Veličković et al. 2020b):

$$h'_w = \text{MLP}_{\theta_4}\left((1 + \epsilon) \cdot h''_w + \max_{w' \in N(w)} \text{MLP}_{\theta_3}\left(h''_w || h''_{w'} || h''_{w,w'}\right)\right)$$

To update the state of an edge, we incorporate its current state along with the updated states of the two nodes it connects.

$$h'_{u,v} = \text{MLP}_{\psi_3}\left(h''_{u,v} || h'_u || h'_v\right)$$

To ensure that the edge remains agnostic to the edge direction and preserve symmetry in undirected edge problems an additional aggregation step is performed:

$$h'_{u,v} = \max\left\{\text{MLP}_{\psi_3}\left(h''_{u,v} || h'_u || h'_v\right), \text{MLP}_{\psi_3}\left(h''_{v,u} || h'_v || h'_u\right)\right\}$$

Finally, the decoder consists of two MLPs that transform the $d$-dimensional embeddings into $|Q|$ node logits, and $|P|$-sized edge logits from which the next states can be sampled:

$$l_w = \text{MLP}_{\theta_5}\left(h'_w\right) \qquad l_{u,v} = \text{MLP}_{\psi_4}\left(h'_{u,v}\right)$$

We use the same architecture for all LCL problems that we consider in the following.

## Experiments

To test our proposed approach across four well-known LCL problems: Maximal Independent Set (MIS), Minimal Vertex Cover (MVC), Maximal Matching (MM), and Minimal Edge Cover (MEC). Both MIS and MVC are node tasks, whereas MM and MEC are edge-centric. For all of them there exists a local verifier that accepts or rejects a proposed solution. For a formal definition of the problems, we refer to the Appendix.

### Supervised Baselines

To critically assess our VARL approach, we compare it to supervised learning baselines. RL is better suited for problems with multiple valid solutions verified by a local checker. Supervised learning, typically requieres a unique label for each instance but can be adapted to handle non-unique cases. We utilize a supervised approach inspired by Luby's algorithm (Luby 1985), which constructs solutions iteratively by selecting nodes or edges based on local properties. At each timestep, the algorithm is given access to a set of random bits. We adapt the selection and invalidation mechanisms for our tasks to match their specific constraints. We consider two different selection strategies:

- Guided Strategy: The solution is constructed by selecting locally maximal elements from the given random bits. The constructed solution uses the exact same set of random bits that is given during training.

- Unguided Strategy: The solution is again constructed by selecting locally maximal elements from random bits. However, the specific random bits are hidden from the model during training. The model still has access to different random bits during training for tie breaking, however, they are independent of the solution. This setup is closer to what we desire to achieve with RL and should encourage independence from specific solving strategies, allowing the model to learn to solve the problem rather than imitating a given algorithm.

|  |  | MIS | MVC | MAT | MEC |
|---|---|---|---|---|---|
| guided | GIN | 100.0 (±0.1) | 100.0 (±0.1) | 63.2 (±29.6) | 74.1 (±10.9) |
|  | GAT | 1.7 (±1.1) | 3.1 (±3.1) | 11.6 (±10.0) | 0.5 (±0.4) |
|  | SAGE | 8.0 (±1.4) | 8.7 (±2.0) | 0.0 (±0.1) | 0.1 (±0.1) |
|  | PGN | 100.0 (±0.0) | 100.0 (±0.0) | 0.0 (±0.1) | 0.0 (±0.1) |
|  | gGCN | 96.1 (±1.1) | 97.4 (±0.8) | 38.9 (±35.2) | 27.6 (±5.7) |
| unguided | GIN | 28.3 (±1.4) | 25.4 (±4.3) | 0.0 (±0.0) | 0.0 (±0.1) |
|  | GAT | 1.4 (±0.7) | 1.1 (±0.1) | 0.0 (±0.0) | 0.0 (±0.1) |
|  | SAGE | 0.9 (±0.2) | 1.0 (±0.4) | 0.0 (±0.0) | 0.0 (±0.1) |
|  | PGN | 25.1 (±4.0) | 26.1 (±2.9) | 0.0 (±0.1) | 0.0 (±0.1) |
|  | gGCN | 22.9 (±2.6) | 20.7 (±4.8) | 0.1 (±0.1) | 0.1 (±0.1) |
| VARL (ours) | large | 100.0 (±0.0) | 100.0 (±0.0) | 99.0 (±0.4) | 98.4 (±0.9) |
|  | small | 100.0 (±0.0) | 100.0 (±0.0) | 99.4 (±0.3) | 99.2 (±0.5) |

|  |  | MIS | MVC | MAT | MEC |
|---|---|---|---|---|---|
| guided | GIN | 100.0 (±0.0) | 100.0 (±0.0) | 95.6 (±5.0) | 98.0 (±1.0) |
|  | GAT | 42.8 (±5.7) | 59.9 (±14.6) | 83.0 (±10.0) | 64.6 (±2.5) |
|  | SAGE | 76.6 (±1.4) | 77.3 (±0.1) | 66.2 (±1.0) | 55.2 (±1.9) |
|  | PGN | 100.0 (±0.0) | 100.0 (±0.0) | 62.1 (±1.0) | 46.7 (±1.9) |
|  | gGCN | 99.4 (±0.2) | 99.6 (±0.1) | 91.2 (±6.5) | 90.1 (±2.2) |
| unguided | GIN | 86.8 (±0.6) | 85.8 (±1.8) | 63.3 (±2.2) | 46.2 (±1.7) |
|  | GAT | 41.0 (±2.8) | 42.2 (±4.7) | 58.1 (±2.7) | 46.2 (±3.4) |
|  | SAGE | 64.2 (±1.0) | 63.8 (±1.1) | 57.0 (±0.4) | 46.1 (±2.4) |
|  | PGN | 85.2 (±1.2) | 86.1 (±1.1) | 61.5 (±2.4) | 46.6 (±1.3) |
|  | gGCN | 83.6 (±1.6) | 83.2 (±1.9) | 66.0 (±1.8) | 53.7 (±1.4) |
| VARL (ours) | large | 100.0 (±0.0) | 100.0 (±0.0) | 99.9 (±0.0) | 99.9 (±0.0) |
|  | small | 100.0 (±0.0) | 100.0 (±0.0) | 100.0 (±0.0) | 100.0 (±0.0) |

Table 1: Our RL-based approach outperforms alternative supervised baselines. We show graph accuracy on top in percentages, and agent accuracy on the bottom. The shown numbers represent mean and sample standard deviation in parenthesis. Rows indicate different methods: supervised baselines have access to labels computed through guided (guided) or unguided (unguided) selection strategy. The reinforcement learning approach is denoted with RL, and we show results from agents with only access to a few states (small), five central and one non-central state, or ones with ten central and five non-central states (large).

The invalidation step ensures solution constraints are satisfied. Using the described strategies, we generate labeled instances for supervised learning datasets formulating the problem as a classification task. Note that there are no specific input features for the nodes as the problems are defined on the graph topology. However, during the execution, the models are given access to random bits. We use different common GNN architectures such as GIN (Xu et al. 2019), GAT (Veličković et al. 2018), SAGE (Hamilton, Ying, and Leskovec 2018), PGN (Veličković et al. 2020a) and Gated GCN (Bresson and Laurent 2018). In order to run them on graphs of variable sizes we set the number of convolutions proportional to the graph size. Furthermore, we also incorporate a memory cell to feed in the random bits and we make the architectures recurrent so that the different convolution layers share the same set of weights.

## Results

We evaluate the discussed baseline models on the four LCL problems and train them on graphs of size 16. In Table 1 we report the agent level accuracy, which indicates the percentage of nodes which the verifier outputs Yes and the graph level accuracy, the number of correctly solved instances.

We can observe large differences between the baselines.

Using labels computed using the unguided strategy, the approach designed to learn to solve the problem rather, is more challenging than the maximum selection, which corresponds to learning a specific algorithm. This indicates that a supervised approach struggles to learn the more general underlying concepts that define solutions from the more general dataset. The used graph convolution layer is also of importance: GAT and SAGE are poor choices, and so is PGN when edge states are central to the problem at hand. SAGE does not consider edge features and uses mean aggregation during message passing, which we found to perform worse when trying different designs for the default modified GIN version. PGN's stark difference between node and edge task is somewhat surprising as edge features are propagated together with node features during message passing. The main difference to our modified GIN concerning message construction is that node and edge features are added together. GIN performs the best across the board. Architectures using gGCN show partial success on edge tasks but perform worse than PGN on node-centric problems.

We test two different versions of our proposed method that is trained using RL. The first variant uses only a few states — if it is used on a node task, $|Q| = 5$ and $|P| = 1$, and on edge tasks $|Q| = 1$ and $|P| = 5$. The second variant, uses 10 and 5 states instead respectively. Both variants are able to achieve very good performance across all tasks with only marginal difference between them, although going with fewer states seems to be slightly better. However, the difference with respect to the supervised baselines is much more significant, especially when considering the number of correctly solved instances. Our proposed method using RL clearly outperforms both supervised strategies, even though it only access to a verifier and has no access to labels.

## Conclusions

Learning correct algorithms purely from data driven feedback is very challenging, especially when no intermediate trajectories by a ground truth mechanism are given or the solution to a given problem instance is not unique. We propose to address these gaps by teaching machines algorithmic thinking through reinforcement learning (RL) for graph problems. We extend the state-based formulation of Graph-FSA to fit within a multi-agent RL framework, where agents on graph nodes observe local states and perform transitions. This has the advantage of generalizing the state updates and also incorporating random bits for necessary tie-breaking. Policies trained via policy-gradient methods are translated into transition functions, modeling the learned algorithm's behavior.

Experiments demonstrate the applicability of our approach to locally checkable problems (LCLs) like maximal independent sets and matchings. Unlike supervised methods that require input-output pairs and struggle to effectively learn these tasks, our verifier-based RL approach learns the underlying problem dynamics and can handles non-unique solutions effectively. We thus validate the feasibility of the proposed RL approaches for learning solvers for algorithmic problems, laying a foundation for further research in algorithmic learning on graphs.

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

# Related Work

## Algorithmic Learning

Extrapolation and logical reasoning are considered to be major weaknesses of neural network-based methods today. The field of neural algorithmic learning combines ideas from classical algorithms, which provide strong generalization and runtime guarantees, with machine learning. The goal is to learn the underlying algorithmic principles in a data-driven approach resulting in learned solvers that manifest corresponding, desirable properties.

The term neural algorithmic reasoning comes from a work with the same name by (Veličković and Blundell 2021). The authors argue that classical algorithm and deep learning approaches fundamentally differ in their generalization capabilities. Many common algorithms find applications in different domains, while deploying neural-network-based approaches in situations that diverge from what was seen in training, oftentimes leads to unpredictable behavior.

We are interested in solving graph problems through state-based methods. In that vein, finite state automata (FSA) and cellular automata (CA) form computation models based on discrete states, which can be used to encode the structure of certain algorithms in a learnable fashion. The earliest example that combines machine learning and CA comes from (Wulff and Hertz 1992). The authors learn the state transitions of simple one and two-dimensional automata with neural networks. More recently, (Mordvintsev et al. 2020) have utilized convolution to learn CA transition rules that generate images from a single seed pixel.

More concretely aimed at solving graph problems is the work that provided the initial inspiration for this thesis by (Grötschla et al. 2024). They build on top of concepts from CA, to develop GraphFSA: a framework for algorithmic learning of problems on general graphs. GraphFSA is based on the observation that some graph algorithms can be modeled by each node following a simple automaton-like state recipe. The GraphFSA's main algorithmic component is the transition function, which guides the nodes' state changes. The authors show that given a dataset of pairs of input and output graphs labeled with states such a function can be learned by back-propagation through a transition table. Their approach shows great extrapolation capabilities and interpretability. However, it is limited by the need for complete discreteness. (Rodionov and Prokhorenkova 2024) relax that limitation in their work called Discrete Neural Algorithmic Reasoning (DNAR). While not explicitly referencing automata, they still rely on discrete state transitions to encode algorithm behavior. However, they make use of continuous inputs and additional edge states to train transitions mimicking more well-known algorithms, such as Luby's algorithm for finding a maximal independent set (Luby 1985), in a supervised fashion.

More concerned about scaling applications and extrapolation results to much bigger graphs than adhering to a discrete state representation is the work by (Grötschla, Mathys, and Wattenhofer 2022). Their experiments on up to 10'000 node graphs show that such extreme size extrapolation is possible yet challenging.

Another work that focuses on extrapolation comes from (Selsam et al. 2019). They present a graph neural network architecture designed with the goal of solving propositional satisfiability (SAT) problems in mind. It shares some similarities with the previous work in that scaling inference to much bigger inputs is facilitated by having an architecture with a variable number of message-passing iterations. This is also a key component in the work by (Tang et al. 2020), who show not only improvement in extrapolation with respect to problem size but also with respect to other factors, such as different edge weight ranges for finding the shortest path in a graph. (Selsam et al. 2019)'s extrapolation demonstration not only includes scaling problem instance sizes either. They also generalize across different problem classes, i.e. their solver is trained on datasets containing randomly generated SAT problems but is able to find solutions for more specific problems such as detecting cliques in a graph.

Algorithmic reasoning has gained more interest in recent years. In order to unify research efforts, (Veličković et al. 2022) have proposed a benchmark suite consisting of multiple different algorithms, input graphs, and corresponding outputs. SALSA-CLRS (Minder et al. 2023) extends that collection by six graph algorithms and exemplifies an experimental methodology that highlights the importance of dataset diversity.

## Reinforcement Learning

Using reinforcement learning (RL) to solve tasks that involve logical reasoning is not uncommon. Probably the most well-known RL-based work, AlphaGo (Silver et al. 2016), involves training an agent to compete in the world of Go, a complex strategy board game.

Applications over graphs are oftentimes looked at through the lens of combinatorial optimization and are more centered around the problem statement than known solving algorithms. For example, (Tönshoff et al. 2022) proposes an approach viable for any constraint satisfaction problem such as MAXCUT or 3-SAT. Formulating the satisfaction problem in the form of a graph allows the authors to train an RL-based search heuristic parametrized as a GNN, whose inherent parallelism can be exploited to speed up training.

The work by (Joshi et al. 2022) also combines GNNs and RL. The authors focus on the traveling salesperson problem (TSP). A comparison between a supervised approach and an RL formulation supports that the latter is a viable alternative and that it can even outperform the former. Related to our work is also the author's focus on generalization across different graph sizes. They identify RL algorithms as one possible key ingredient next to inductive bias for extrapolation to very large graphs.

Other famously hard problems are also studied. E.g. (Huang, Patwary, and Diamos 2019) investigate coloring problems. A rigorous survey of works that combine RL with combinatorial optimization is provided by (Mazyavkina et al. 2020), or more recently with a focus on GNNs by (Cappart et al. 2022).

This thesis will formulate learning graph algorithms as a multi-agent reinforcement learning (MARL) task where the number of agents scales with the size of the graph. A

method of keeping the number of trainable parameters manageable even under such circumstances is using shared policies. (Christianos et al. 2021) show that parameter-shared approaches are capable of solving well-known RL sample tasks such as Level-based Foraging and Starcraft Multi-Agent Challenge.

The theoretical justifications for policy sharing are provided by (Terry et al. 2023). In their work, the authors prove that agents that are aware of a unique identifier and operate under parameter sharing may converge to the optimal multi-agent policy.

Finally, extensive benchmarking of MARL learning algorithms by (Papoudakis et al. 2021) shows that shared policies can outperform other options when combined with independent learning and trained through policy-gradient methods; even while operating under sparse reward signals.

## Multi-Agent Reinforcement Learning

Many applications can not be modeled with a single agent. Allowing $k$ agents leads to the notion of multi-agent reinforcement learning (MARL). The above definition of MDPs can be adjusted accordingly: each agent $i$ operates over its own action space $\mathcal{A}_i$, is assigned its own reward signal $\mathcal{R}_i : \mathcal{S} \times \mathcal{A}_i \mapsto \mathbb{R}$, and the transition function's domain changes to $\mathcal{S} \times \vec{\mathcal{A}} \times \mathcal{S}$, where $\vec{\mathcal{A}}$ is the *joint-action space* $\mathcal{A}_1 \times \mathcal{A}_2 \times \cdots \times \mathcal{A}_k$.

We can also extend the notion of policies to the multi-agent scenario: policy $\pi_i$ maps $a_j \in \mathcal{A}_i$ given $s \in \mathcal{S}$ to the probability of agent $i$ taking action $j$ when in state $s$. Alternatively, a joint-policy $\vec{\pi}$ models the behavior of all the agents in parallel: It assigns a probability to a joint action $\vec{a} \in \vec{\mathcal{A}}$ given the environment's state. However, *central learning* (Albrecht, Christianos, and Schäfer 2023) — a MARL approach that operates on joint policies — faces the problem that the joint action space size grows exponentially for each added agent, i.e.: $|\vec{\mathcal{A}}| = |\mathcal{A}_1| \cdot |\mathcal{A}_2| \cdot \cdots \cdot |\mathcal{A}_k|$.

A method that avoids this quickly growing joint-action space is called *independent learning* (Albrecht, Christianos, and Schäfer 2023). Each agent $i$ is modeled separately in a way that assumes all other agents as part of the environment. For an agent, this assumption may make the transition function appear non-markovian as other agent's behavior may change throughout training. However, it also allows the use of single-agent RL algorithms.

# Problem Formulation

## Training Procedure

One major specification of our state-based approach significantly shapes the proposed procedure of this work: all nodes follow the *same* state transition recipe. In other words, two nodes in the same state, both having the same distribution of states in their immediate neighborhood must transition to the same state. The transition function $T$ was therefore defined irrespective of nodes. The formulation of the multi-agent learning task — an agent is placed on each node, and the transition function is derived from the agent's policy — implies that if we want the invariance across nodes to hold, $\pi_i$ should be equal to $\pi_j$ for every pair of trained

agents $i, j$. Luckily, this can be enforced through parameter sharing. Namely, for two agent $i, j$ with their policies modelled by $\pi_i(\cdot|\cdot, \theta_i)$ and $\pi_j(\cdot|\cdot, \theta_j)$ respectively parameter sharing implies $\theta_i = \theta_j$. Our proposed independent learning MARL training procedure with shared policies will use policy-gradient learning algorithms.

A trained policy needs to properly address that a node is not allowed to switch away from a final state. Putting it into reinforcement learning terms, we want to restrict a trained agent's actions if certain conditions are met. We implement this as follows: We mask logits associated with unwanted actions. Let $\vec{l} = (l_1, l_2, \ldots, l_{|\mathcal{A}|})$ be the output of the shared policy network of some agent whose node is in state $q_i$. The corresponding masked logits $\vec{l'} = (l'_1, l'_2, \ldots, l'_{|\mathcal{A}|})$ are defined as:

$$l'_j = \begin{cases} l_j, & \text{if } q_i \notin Q_F \text{ or } i = j \\ -\eta, & \text{if } q_i \in Q_F \text{ and } i \neq j \end{cases}$$

Choosing $\eta$ to be sufficiently large and defining the policy as the softmax over masked logits:

$$\pi(a_j|o, \theta) = \frac{exp(l'_j)}{\sum_{a_k \in \mathcal{A}} exp(l'_k)}$$

means that transition probabilities attain close-to-extreme values if $q_i$ is a final state:

$$\pi(a_j|o, \theta) \approx \begin{cases} 0, & \text{if } i \neq j \\ 1, & \text{if } i = j \end{cases}$$

With this, masking does not only disallow the transition function defined over the argmax of the policy to switch away from a final state but also makes it effectively impossible to sample such illegal transitions while sampling from $\pi$ during training. Preliminary experiments showed that explicit masking leads to better results than implicitly discouraging unwanted outbound transition by penalization through the reward. The choice is further supported by (Huang and Ontañón 2020)'s work about invalid action masking, which tells a similar story, and provides a theoretical justification for action masking in general.

## Reinforcement Learning Formulation

Throughout the execution of an algorithm in our state-based framework, $v$ will transition from state to state. Say the execution takes $t$ steps, and $v$ encounters states $q_0, q_1, \ldots, q_t$ in that order. In each state $q_i$, node $v$ applies $T$ to get $q_{i+1}$. This iterative application of a transition function parallels an agent's trajectory in a MARL setting if the action it can take corresponds to algorithm-state transitions: the node's agent started in state $q_0$, it took the action corresponding to transitioning to state $q_1$, then the action to go to $q_2$, and so on. Zooming out on the whole graph, placing an agent on each node, and allowing it only to observe its neighbors' states, results in an intuitively analogous MARL formulation.

To combat possible confusion during the merging of the two concepts with similar terminology, the congruence is shown here more formally. The following deterministic

multi-agent MDP corresponds to an environment for an algorithm solving some problem $\mathcal{P}$ using states $Q, Q_0, Q_F$ as previously introduced: an MDP state $s$ from state space $\mathcal{S} = Q \times Q \times \cdots \times Q = Q^n$ at time step $t$ includes all the $n$ nodes' algorithm states, i.e. $s = (q^{v_1}, q^{v_2}, \ldots, q^{v_n})$. All $n$ agents share the same action space $\mathcal{A} = \{a_0, a_1, \ldots, a_{|Q|-1}\}$, where $a_i$ implies the agent's node swapping to state $q_i$. Therefore:

$$\mathcal{T}(s, \vec{a}, s') = \mathcal{T}\big((q^{v_1}, \ldots, q^{v_n}), (a^{v_1}, \ldots, a^{v_n}), (q'^{v_1}, \ldots, q'^{v_n})\big)$$
$$= \begin{cases} 1, & \text{if } \forall v \, \exists i \text{ s.t. } a^v = a_i \wedge q'^v = q_i \\ & \wedge (q^v \in Q_F \Rightarrow q^v = q_i), \\ 0, & \text{otherwise.} \end{cases}$$

Note that the number of environment states grows exponentially in $n$, which is exacerbated by higher numbers of node states. We have that $|\mathcal{S}| = |Q|^n$. However, the restriction through final states can limit the number of valid environment transitions.

The only thing missing from the MARL formulation is the reward function. We have neither access to a ground-truth transition function nor any hints that can be computed through it. Yet, a verifier is available. For a node-based verifier $\mathcal{V}$ we define the reward for an action of agent $i$ located on node $v_i$ in the environment state $s_t$:

$$\mathcal{R}_i(s_t, a) = \begin{cases} 1, & \text{if episode end} = t + 1 \text{ and } \mathcal{V}(v_i) = \text{Yes} \\ -1, & \text{if episode end} = t + 1 \text{ and } \mathcal{V}(v_i) = \text{No} \\ 0, & \text{otherwise} \end{cases}$$

Or analogously, the reward of agent $j$ associated with edge $e_j$ if $\mathcal{P}$ is an edge-centric problem:

$$\mathcal{R}_j(s_t, a) = \begin{cases} 1, & \text{if episode end} = t + 1 \text{ and } \mathcal{V}(e_j) = \text{Yes} \\ -1, & \text{if episode end} = t + 1 \text{ and } \mathcal{V}(e_j) = \text{No} \\ 0, & \text{otherwise} \end{cases}$$

An episode of a node-centric task is considered finished if all nodes reach a final state or the horizon is reached. For an edge-based task, we consider the final edge states for the termination condition.

The environment state in this MARL setting with node and edge agents contains all their respective node or edge states, which if they are final, encode matching output labels. In the last environment state $s_{t+1}$ of the episode, the verifier is executed, and based on its output, a non-zero reward is paid. If the episode is preempted due to reaching the horizon, some node or edge state may be non-final. In that case, the corresponding agent just receives $-1$ reward and its state is interpreted as an additional label for its neighbor verification computation.

## Problem Definitions

Our proposed approach is tested on four problems. As they are not part of a preexisting benchmark as was the case in the previous chapter, we introduce them more formally here. The objective of all four of them is to find either a set of nodes or a set of edges for a given graph. The four sets of interest are defined as follows:

**Definition 1 (Maximal Independent Set)** *A set of vertices $S \subseteq V$ is an independent set if no node $v \in S$ has a neighbor that is also in $S$. An independent set $S$ is maximal if, for every node $u$ not in $S$, $S \cup \{u\}$ is not an independent set.*

**Definition 2 (Minimal Vertex Cover)** *A set of vertices $S \subseteq V$ is a vertex cover if every edge $e \in E$ has at least one endpoint in $S$. A vertex cover $S$ is minimal if for every node $v$ in $S$, $S \setminus \{v\}$ is not a vertex cover.*

**Definition 3 (Maximal Matching)** *A set of edges $M \subseteq V$ is a matching if every node $v \in V$ is incident to at most one edge in $M$. A matching $M$ is maximal if for every edge $e$ not in $M$, $M \cup \{e\}$ is not a matching.*

**Definition 4 (Minimal Edge Cover)** *A set of edges $S \subseteq E$ is an edge cover if every node $v \in V$ is incident to at least one edge in $S$. An edge cover $S$ is minimal if for every node $e$ in $S$, $S \setminus \{e\}$ is not an edge cover.*

With respect to labels, the four problems share similarities. They do not come with inputs, i.e. the input alphabet $\Sigma_{\text{in}}$ only contains an empty label $\sigma_\lambda$, and outputs can be encoded as simple membership indicator $\sigma_0 \in \Sigma_{\text{out}}$ represents a node or an edge being in the solution set, $\sigma_1 \in \Sigma_{\text{out}}$ if it is not. Yet, they differ in other aspects. maximal independent set (MIS) and minimal vertex cover (MVC) are both node-centric tasks. On the other hand, the goal of maximal matching (MAT) and minimal edge cover (MEC) is to extract an edge set, and as such labels are associated with edges. Orthogonally to that, as is already implied by the names, MVC and MEC are minimality problems, while MIS and MAT concern themselves with maximality.

## Experimental Setup

The policies were trained with the REINFORCE (Williams 1992) algorithm using a batch size of 16, discount factor $\gamma = 0.95$ and an entropy coefficient of $10^{-6}$. The Horizon $H$ was chosen to be roughly $4 \log_2 n$. Throughout the experiment, we use the Adam optimizer (Kingma and Ba 2017) with a learning rate of 0.0003. The supervised baselines perform 160 epochs during training, with a batch size of 16 and the train set containing $10,000$ graphs equivalent to $100,000$ optimization steps. We fix the number of recurrent updates, to be $4 \cdot (\log n + 1)$, which limits the supervised baselines to use the same upper bound for the number of communication rounds as the RL agents. Model selection is based on the best parameters found during training on the validation set.