# OpenReview forum: "Reinforcement Learning for Locally Checkable Labeling Problems"
_AAAI.org/2025/Workshop/NeurMAD — AAAI 2025 Workshop NeurMAD Submission_

### Official Review · Reviewer_ZdPW · 2024-12-18
**Application of reinforcement learning framework for Locally Checkable Labeling (LCL) problems.**

**Rating:** 7
**Confidence:** 4

**Review:**

**Paper summary**
VARL is a reinforcement learning-based framework designed to solve Locally Checkable Labeling (LCL) problems. Compared to traditional supervised learning methods, this approach eliminates the reliance on ground truth solutions and avoids bias toward specific algorithms. By leveraging local validators to evaluate solutions, it is not constrained to unique solutions and can effectively handle tasks with multiple valid solutions. It has demonstrated effective generalization capabilities across multiple tasks, outperforming supervised learning baselines.
**Originality**
- **Strengths**: VARL follows the framework of the supervised method GraphFSA but relaxes the constraints on the transition function, making it more generalized. Each node and edge is treated as an agent that independently selects its next state, while all nodes or edges share the same policy, effectively reducing the number of parameters. The rewards are from the results of local validators and guide the model to find valid solutions.
- **Weaknesses**: While the idea of allowing the model to freely explore all possible solutions through the reinforcement learning process is interesting, its time complexity needs further consideration. The approach of exploring all valid solutions without relying on specific algorithms could be seen as a heuristic. However, the exploration process in reinforcement learning, how to avoid redundant computations, and how to ensure all solutions are effectively discovered require deeper investigation.

**Quality**
- **Strengths**:   The technical derivations are sound. The model employs a multi-agent reinforcement learning framework, graph convolution layers for aggregating neighbor states, and MLPs for encoding and decoding. The performance improvements demonstrated in the experiments support this approach.
- **Weaknesses**: This reinforcement learning-based approach does not rely on ground truth or specific algorithms, which might necessitate a complete exploration of the entire graph. The paper lacks a theoretical analysis of the time complexity, and the experiments are conducted solely on small graphs of size 16, without extending to larger graphs. Furthermore, the paper does not provide detailed descriptions of the design and computational cost of the local validator. In theory, the validator would need to be manually designed for different tasks.

**Significance**
- **Strengths**:   This method differs from traditional supervised methods as it does not rely on ground truth or specific algorithms, yet achieves better performance across various tasks, demonstrating strong generalization capabilities. It provides a universal foundation for learning algorithmic reasoning on graphs. The experimental results indicate that the combination of reinforcement learning and local validators offers a promising direction for addressing algorithmic problems on graphs, with broader implications for learning and reasoning in discrete domains.
- **Weaknesses**:   Although the results are promising, the time complexity is not analyzed, making it difficult to assess the practical applicability of the approach. The broader applicability of this method has not been thoroughly discussed. Incorporating more comprehensive baselines or complexity comparisons would further enhance its significance.

**Questions and suggestions for the authors**
- In addition to supervised learning baselines, could the authors consider introducing other reinforcement learning methods for comparison to more comprehensively evaluate the performance advantages of the proposed approach. Furthermore, incorporating traditional heuristic-based algorithms as additional baselines, particularly well-known classical algorithms in LCL problems, could provide valuable insights.

- Testing on larger-scale graph structures (e.g., with 100+ nodes or more) would help analyze changes in training efficiency and validator performance. Additionally, evaluating the scalability of the proposed method in real-world large-scale network datasets would strengthen its practical applicability.

- Could the authors provide detailed descriptions of the local validators, such as pseudocode or process flows for each task, and analyze their time complexity to give a clearer understanding of their computational requirements.

**Limitations**
The authors repeatedly emphasize that the local validator is a core component, but they do not provide detailed descriptions of its design or computational cost. For different tasks, this module may require manual design, which could limit its general applicability. Another limitation is that, although the model demonstrates strong performance and generalization capabilities, its complexity and training time costs cannot be adequately evaluated. Additionally, the correctness of the solutions relies entirely on the validator, without robust theoretical guarantees to support its reliability.

**Ethics**
There are no obvious direct ethical concerns related to the method as it stands. The paper does not deal with sensitive data or produce sensitive content. The approach is a method improvement and not directly involved in human-facing decision-making applications at the evaluation stage. No unethical dataset or methodology usage is apparent. Thus, no ethical issues need to be flagged for special ethics review.

---

### Official Review · Reviewer_JAob · 2024-12-19
**The paper proposes a novel RL framework for LCL problems and outperforms supervised baselines.**

**Rating:** 6
**Confidence:** 3

**Review:**

Summary
The paper proposes an RL framework for solving LCL problems on graphs, leveraging local verifiers instead of ground-truth labels. This approach avoids algorithmic biases and supports non-unique solutions. The proposed framework significantly outperforms supervised baselines on four LCL problems.

Strengths
- The writing is clear and well-structured.
- The proposed framework achieves impressive performance across diverse LCL problems, surpassing supervised methods by large margins.
- The proposed framework is novel and does not require ground-truth labels or pre-defined algorithms.

Weaknesses
- In the experiments, does the VARL model have a comparable number of parameters to the baseline models? It is unclear how much of the observed improvement is due to having more parameters.

Suggestions
- It would be better to describe your reinforcement learning algorithm in the main text rather than in the appendix, as the paper proposes a reinforcement learning framework.
- The first sentence of the Experiments section, "To test our ...", should be revised to "We test our ..." for grammatical correctness.
- It would be better to evaluate the proposed framework on large-scale datasets.

---

### Official Review · Reviewer_Gkbx · 2024-12-28
**A Novel Multi-Agent Reinforcement Learning (MARL) Framework for Locally Checkable Labeling (LCL) Problems**

**Rating:** 7
**Confidence:** 3

**Review:**

Summary:
The authors tackle the problem of solving locally checkable labeling (LCL) tasks on graphs using Multi-Agent Reinforcement Learning framework. In contrast to traditional supervised approaches that rely on ground-truth algorithms or enforce unique solutions, their framework uses verifiers to evaluate correctness. This method allows models to learn solution strategies independently, free from biases toward specific algorithms and supports the discovery of multiple valid solutions. They validate their framework on four core LCL problems, showcasing its ability to generalize effectively, surpass supervised baselines, and provide a flexible foundation for algorithmic reasoning on graphs.

Key Contributions:
* RL-based Framework: The method trains agents (representing nodes or edges) to learn decision-making policies based on local observations and verifiers.
* Flexibility: It supports problems with multiple valid solutions and avoids biases tied to specific algorithms.

The framework was tested on the four LCL problems:
* Maximal Independent Set (MIS)
* Minimal Vertex Cover (MVC)
* Maximal Matching (MM)
* Minimal Edge Cover (MEC) Results demonstrate superior performance compared to supervised baselines, particularly in generalization and solving edge-centric problems.

Strengths:
* The paper is clearly and effectively written.
* The proposed framework achieves better performance compared to supervised baseline methods.
* The RL-based approach eliminates the need for predefined labels or unique solutions, making it highly adaptable to a variety of LCL problems.
* The framework effectively handles problems with multiple valid solutions, addressing a significant limitation of many supervised methods.

Limitations:
* The RL-based framework involves tuning multiple parameters (e.g., reward functions, policy networks), which can lead to increased computational complexity.
* It is recommended to move the description of the RL framework from the appendix to the main paper, as it represents the core contribution.
* Clarification is needed regarding the dataset generation process for training, validation, and testing. Were the graphs generated randomly, and what is their distribution?
* Testing the framework on larger graphs with more than 16 vertices would have provided a more comprehensive evaluation.

Typos:
Page 2: “In GraphFSA, ach node...” change ach to each.
Page 4: “Architectures using gGCN show partial success...” change gGCN to GCN

---

### Decision · Program_Chairs · 2024-12-30

**Decision:**

Reject

**Comment:**

This is a good paper. However, it does not fit the scope of this workshop.